# Expressing History through a Geo-Spatial Ontology

## Humphrey Southall *[ID] and Paula Aucott[ID]

Department of Geography, Buckingham building, Lion Terrace, University of Portsmouth,
Portsmouth PO1 3HE, UK
* Correspondence: humphrey.southall@port.ac.uk

**Abstract:** Conventional Geographical Information Systems (GIS) software struggles to represent uncertain and contested historical knowledge. An ontology, meaning a semantic structure defining named entities, and explicit and typed relationships, can be constructed in the absence of locational data, and spatial objects can be attached to this structure if and when they become available. We describe the overall architecture of the Great Britain Historical GIS, and the PastPlace Administrative Unit Ontology that forms its core. Then, we show how particular historical geographies can be represented within this architecture through two case studies, both emphasizing entity definition and especially the application of a multi-level typology, in which each "unit" has an unchanging "type" but also a time-variant "status". The first includes the linked systems of Poor Law unions and registration districts in 19th century England and Wales, in which most but not all unions and districts were coterminous. The second case study includes the international system of nation-states, in which most units do not appear from nothing, but rather gain or lose independence. We show that a relatively simple data model is able to represent much historical complexity.

**Keywords:** administrative units; ontology; historical geography; England and Wales; Poor Law unions; registration districts; countries; borders

## 1. Introduction

History is concerned not just with the past but specifically with the interpretation of documentary sources, distinguishing it from archaeology. Some historical documents are maps, but their depiction of relative locations cannot be assumed to be accurate. The vast majority of historical documents include neither maps nor numerical coordinates, so "geographical information" exists almost entirely through the inclusion of geographical names, or toponyms. Applying Geographical Information Systems (hereafter 'GIS')-based methods historically therefore generally requires the construction of knowledge bases enabling the translation of toponyms, as they appear in historical documents, into geographical coordinates.

The simplest form of such knowledge bases is the classic gazetteer, such as the NGA Gazetteer (http://geonames.nga.mil/gns/html/) or Geonames (https://www.geonames.org/): simply a list of toponyms each with a corresponding coordinate, plus often a "feature type" such as "water feature" or "populated place". Such gazetteers generally derive directly from topographic mapping, with feature typing derived from map symbologies. However, they are often poorly suited to historical use. One problem is that toponyms rarely work as unique identifiers, so a particular name may refer to many different locations, and a particular location will commonly have multiple names, sometimes in multiple languages and changing in importance over time. More complex structures are clearly required, with multiple toponyms per feature, detailed source information for each toponym and, to disambiguate common names, some notion of hierarchy. For example, one "Newport" is in the county of Shropshire, another "Newport" is in Monmouthshire (or, alternatively, Casnewydd is in Gwent, as this second Newport is in Wales).

Topographic maps seek to capture the physical landscape and deal primarily with 'features' with a physical existence. However, the toponyms in historical documents rarely refer to objects that we can touch, but instead either to more vaguely defined 'places', which are often but not always settlements, or to legally defined administrative areas, ranging from the kingdoms and republics that are the stuff of political history, through the diverse provinces, counties, and municipalities listed in census reports, to the parishes that were the units of record for births, marriages, and deaths for most of the last 500 years in Europe. A specifically historical agenda for gazetteer research was set out by Southall et al. [1], while Berman et al. [2] provides a systematic survey of historical gazetteer activity.

'Places' have been the main focus of most recent work, and are best conceived as fuzzy polygons. Instead, this paper is concerned mainly with administrative areas, or 'units' that we generally assume to have had precise, legally defined boundaries; however, representing historical units within a GIS faces a series of challenges. Firstly, while digital boundaries for modern units are usually available from government, these generally cover only the last 30 to 50 years, and while very detailed paper boundary mapping is often available for the last 100 or more years, digitization is very time consuming and expensive. Secondly, boundaries may have been well-defined in periods well before they were mapped, but either no record exists at all, or there is only a textual description, from which it is even harder to construct a polygon; the US Atlas of Historical County Boundaries was constructed this way, but took decades [3]. Thirdly, as they go back further in time, historians increasingly encounter units listed in, for example, taxation records, whose geographical location is unknown, other than through their being part of some higher level unit.

For all these reasons, building a knowledge base of historical administrative units as a conventional polygon-centered GIS is inappropriate. Digitization costs alone mean that at least our initial system needs to be based primarily on textual sources, and therefore must be a primarily semantic structure to which we can add geospatial data as it becomes available. There are two broad models of knowledge organization systems for holding such information: thesauri and ontologies. In brief, the former define 'vocabularies' made up of 'terms', which can be 'broader' and 'narrower', defining hierarchies, and 'preferred' and 'alternate', enabling variant names to be included. Ontologies differ firstly by being more concerned with fundamental concepts, and secondly and more practically through a greater concern with relationships. In a "polyhierarchic thesaurus", a narrower term can lie within multiple broader terms, but only in an ontology can there be multiple kinds of relationships [4].

This paper is centrally concerned with the PastPlace Administrative Unit Ontology (hereafter AUO) developed by the Great Britain Historical GIS (hereafter GBH GIS), which seems to be unique in being both historical and multi-national, and unusual in that it frequently includes vector boundary data while requiring only hierarchic relationships. A search interface to the AUO is provided by the Vision of Britain site (http://www.visionofbritain.org.uk/expertsearch), which will be re-named as PastPlace by the end of 2019 (https://www.pastplace.org/expertsearch). The re-naming reflects the system including significant non-British content, as discussed below, and the re-launched system will also include API access at data.pastplace.org.

A number of other projects have taken broadly similar approaches. The United Nations Food and Agriculture Organization's (FAO) Ontology is global but limited to currently existing countries [5], while national ontologies exist for contemporary Britain [6], Turkey [7], and China [8]. Kauppinen et al. developed a pioneering formal "geospatial ontology time series" for Finland, although this is relatively small and makes an arguably problematic distinction between "modern" and "historical" units, rather than all units existing in continuous time [9]. Lacasta et al. constructed a "spatio-temporal knowledge base for jurisdictional domains" for Spain using automated methods by basing the structure on existing listings where start and end dates are already determined, but construct a separate domain for each hierarchy without overlap [10].

The AUO, and the overall historical GIS that it is part of, have already been described at length [11–13], while Aucott et al. documented its extensions beyond Britain [14]. The next section summarizes the database architecture, but at this level, the system is extremely abstract: it defines

entities that can have various names, and have various kinds of relationships with other entities. For example, this structure could just as easily hold information on people and their family trees. Therefore, the main focus here is on the next level of generality, where particular administrative geographies are defined. The presentation is primarily through two detailed cases studies. Section 3 describes the representation of the system of Poor Law unions and registration districts, which were the most important statistical reporting units in late 19th century Britain, and the focus of the earliest work of the GB Historical GIS. Then, the more exploratory Section 4 discusses how best to represent the changing international system of "countries" since the Vienna Congress (1815).

One aim is to show historians how an apparently abstract architecture can in fact accommodate much historical specificity, but another is to explain to information scientists why modeling must include such detail, rather than simplifying administrative geography to just a set of 'levels'. This paper is about the actual knowledge content, not about particular representations. In particular, current work, in collaboration with FreeUK Genealogy and funded by Pelagios Commons, to expose the AUO as Open Linked Data [15] will be reported elsewhere. This work includes a formal representation of the typologies described below using the W3C SKOS (Simple Knowledge Organization System) RDF Schema, which is an OWL ontology [16]. In what follows, we focus on the internal representation of the data within a Postgres object-relational database using the PostGIS spatial extension.

## 2. Overview of the GBH GIS Administrative Unit Ontology

The original GB Historical GIS was developed in 1994–2000 primarily using ArcGIS software with very extensive scripting in the now-obsolete Arc Macro Language. The scripting enabled the system to include dates despite the complete lack of support for a time dimension in the underlying software. The system stored individual boundary segments as arcs, each with a start and end date, and identified administrative units via 'label points', which included a single name and, again, had start and end dates. Then, a more conventional polygon coverage could be constructed for a given date by selecting those label points whose period of existence included the target date, and building polygons around them from the arcs that existed at the target date. Statistical data were held in a separate Oracle database that could be accessed from ArcGIS, and linked to via the names held in the label points [17,18].

One limitation of this architecture was that construction was very time consuming, especially given the need to ensure that a valid polygon topology was in fact buildable for a full range of dates. This meant that GIS construction often lagged far behind statistical transcription, and one major analytic project had to develop a quite different methodology for analyzing mortality decline in 20th century Britain [19,20]. Another limitation was that the GIS held just one name for each unit, while the historical sources often contained valid alternate spellings. This was initially addressed through the construction of 'name standardization tables' within Oracle. Further, even in the earliest work on registration districts as discussed below, there were two districts in England and Wales called "Newport", two called "Richmond", and two called "Wellington". The temporary solution was to include abbreviated county names within brackets in the labels for just these units, but it was increasingly obvious that a more systematic approach was needed for variant and ambiguous unit names.

This problem was shared with the archives sector in the UK, which—unlike libraries—routinely index by place, meaning not some GIS-based method but alphabetic indexing by place name, traditionally using index cards and now in specialized collections management software. The system of local record offices were generally doing this using the name of the parish containing the address, and the National Council on Archives (hereafter NCA) had identified a series of books as "name authorities", providing the standard forms of the names of parishes and other administrative areas [21]. In 2001, the GB Historical GIS received new funding from the UK National Lottery, both to make our statistical data available to the general public, especially as local time series, and to provide an online name authority for the archives sector based on computerizing the existing authorities; meeting both goals with a single system.

The most important authority identified by the NCA was Frederick Youngs' two-volume 'Guide to the Local Administrative Units of England' [22,23], which contain no maps whatsoever. The volumes are organized geographically by county; then, each county is sub-divided into entries for parishes, the most enduring and generally the most detailed administrative geography. Entries list each parish's relationships with generally more ephemeral higher-level administrative units. Figure 1 is a simplified representation of the database structure developed for the AUO, which was originally used to hold data manually extracted from Youngs', and includes all the tables holding information about individual units. In brief, the units table enumerates all the units of all types, but holds little information about them other than dates of creation and abolition, if known. The names table holds all the names that units are or were known by. The relationships table holds hierarchic relationships and also boundary changes, such as "abolished to enlarge"; by definition, any change makes one unit larger and one smaller. The "footprints" table holds boundary polygons. The use of the status table is discussed below. Strictly speaking, our statistical data is distinct from the AUO, but as shown here, all the data values have their own row in a single "data table", which also holds a unit ID number linking it to the units table [11].

All the tables include start and end dates, and all the rows in all the tables are required to include authority identifiers linking to a single central table of authorities: all information must be attributed to a specific source. In most contexts where attributions can be included, the system can identify both an "immediate authority", meaning the source from which we took the information, and an "ultimate authority", meaning a legal instrument such as a decree or treaty actually making the change, as listed in the immediate authority. This detailed documentation of sources is almost impossible in traditional GIS, due to the very widely used Shapefile imposing a 254-character limit on any annotations.

A complete UML diagram for the AUO has been published elsewhere [12], and the system includes two additional types of table, which are both much smaller. Firstly, and as originally constructed to support data from Youngs' and other British sources, separate tables define:

- What types of units can exist.
- What types of relationship can exist.
- What types of relationship can exist between particular pairs of unit types.
- What status values each unit type can have.
- What names statuses, such as 'Preferred' and 'Alternate', may be associated with each name.
- What types of authority can exist, as books such as Youngs' are treated differently from census listings.

Secondly, the only major changes to this architecture since it was defined in 2002–2003 were under the QVIZ project in 2006–2008, when the system was internationalized: an existing table identifying the languages used in names was extended, and several tables added to enable explanatory text to be held in multiple languages [14,24]. This was also when the opportunity was taken to expand the structure from a base root unit of 'the British Isles' up to 'the world', enabling the system to also hold detailed data for Estonia and Sweden, and potentially elsewhere.

As noted above, the database shown in Figure 1 could instead hold information about people: for example, the units table would contain a Social Security number plus dates of birth and death; the names table would hold forenames and family names, and possibly other unique identifiers; relationships would contain, for example, 'child of' and 'married to'; status would hold titles such as 'Mr., Mrs., Miss., or Dr.'; footprints would hold changing addresses. To understand the actual use of the system as an administrative unit ontology, and especially the use of unit types and status, we need to consider specific examples.

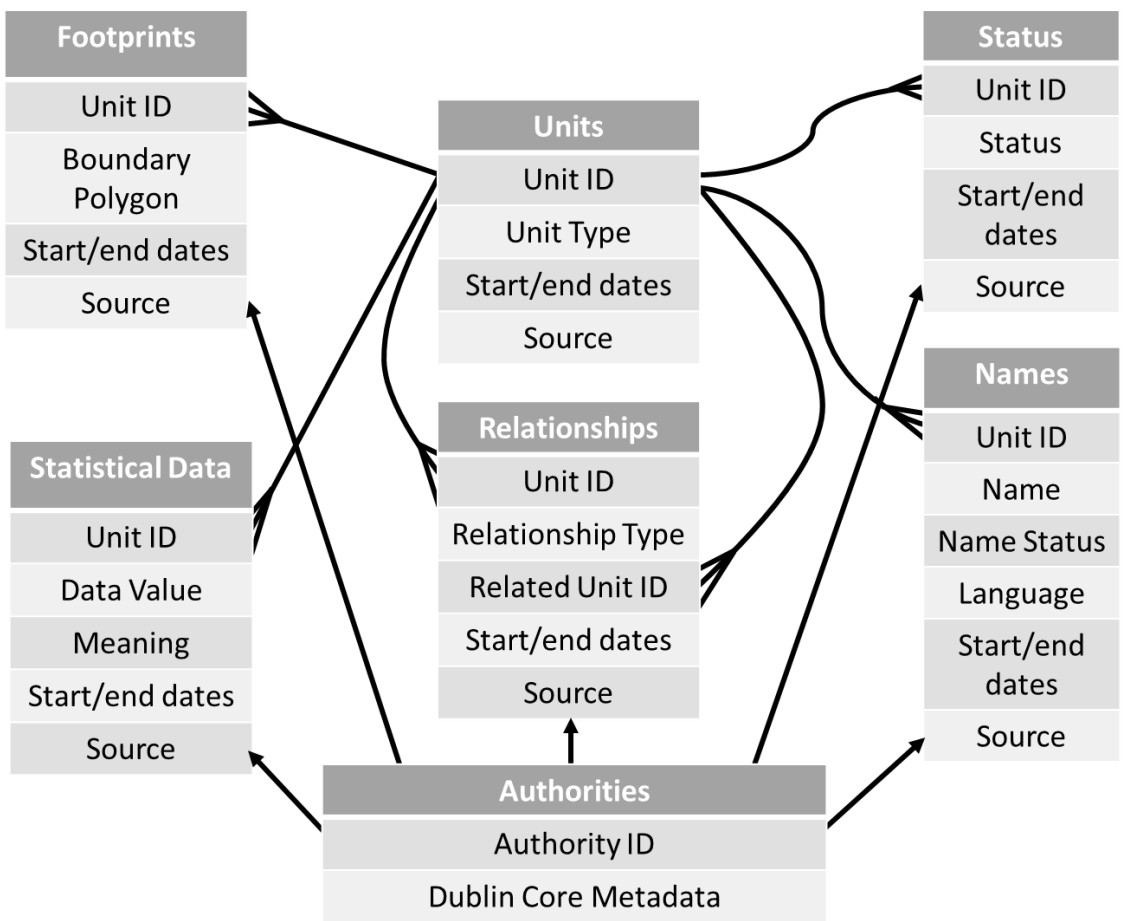

**Figure 1.** Simplified PastPlace Administrative Unit Ontology (AUO) Data Model.

## 3. AUO Unit Type and Status Differentiation: Poor Law Unions and Registration Districts

Poor Law unions replaced the administration of poor relief by individual parishes in the first half of the 19th century. They were created under the Poor Law Amendment Act of 1834 (4 and 5 Will 4, c. 76), and commissioners generally grouped 20–30 parishes within 10 miles of a market town to form a union [25]. Central to the system was the Poor Law Commission, acting as a single authority across the whole country, and replacing the disjointed and self-governing local schemes [26]. However, in practice, the commission was operating a policy incorporating a string of compromises negotiated between different official boards at both national and local levels [27]. This national system governed by a single set of homogeneous rules was enforced locally by a Board of Guardians—or, prior to their establishment, other specific local act bodies until such boards could be created [28].

Initial progress in implementing the new system was quick. During 1834, 2300 parishes were incorporated into the new system, and by the end of 1839, 13,691 parishes and townships (95%) had been brought into the union system, leaving just 799 parishes still operating under their own local authority [25]. However, because the unions were implemented alongside existing incorporations and Gilbert Unions, which were not spatially contiguous, inconsistencies resulted in provision, governance, and reporting in certain areas of the country. Thus, the 1834 Act and subsequent Poor Law policies were shaped by both the existing geography of Poor Law provision and subsequent social development [27]. A series of acts following on from 1834 tried to alleviate the compatibility issues and consolidate these differing local entities into a single system. While the original act was amended several times, reform was never satisfactorily completed, and eventually led to all unions being abolished in 1930 under the Local Government Act of 1929 (19 and 20 Geo V, c. 17), and the subsequent abolition of the Poor Law itself in 1948 under The National Assistance Act.

Much of what we know of the creation and component parishes for Poor Law unions is derived from the lists given in the 1887 'Statement of the names of the several Unions and Poor Law Parishes in England and Wales', together with our main source for constructing the AUO for England Youngs' Guide [22,23]. Youngs' volumes give no detail about boundary changes, other than a year and a reference, usually to the London Gazette. This makes boundary changes over time difficult to document fully. Additionally, the Youngs' books make no mention of the almost equivalent registration districts.

The system of registration districts in England and Wales was similarly imposed by central government in 1837 to record births, marriages, and deaths independently from the Church of England. It was introduced following two 1836 Acts; the Marriage Act (6 and 7 Will. IV, c. 85) and the Births and Deaths Registration Act (6 and 7 Will. IV, c. 86). Registration districts were and are the spatial units for the compulsory recording of all vital events for the entire population by the Registrar General. Additionally, they were used as the primary reporting geography for the Census of Population between 1851–1911. Scotland had a separate system of registration districts introduced in 1855, which is not covered here. The registration districts were introduced along with sub-divisions called 'registration sub-districts', and the districts were aggregated in statistical reports into 'registration counties', which had no administrative function and were identical to the 'union counties' into which Poor Law units were similarly grouped. Each unit was defined primarily in terms of its component parishes, although urban sub-districts could be sub-divisions of parishes. The registration district geography generally followed the Poor Law union geography, enabling the two systems to share buildings and officials, but as discussed below, there were significant differences in particular localities.

Registration districts and Poor Law unions were the very first administrative units to have their boundaries mapped by the GB Historical GIS Project, working in the mid-1990s. Our work developed out of the Labour Markets Database [29], which focused on measures of economic distress, and our immediate aim was not to map census data at 10-yearly intervals, but to map and analyze Poor Law statistics, which are reported twice a year [30], and vital registration data, especially marriage statistics, which are reported annually. Southall and Gilbert [31] showed that fluctuations in the marriage rate varied closely with the state of local economies. Requiring mapping for so many different dates justified the development of a unique time-variant architecture [18], rather than following earlier projects in constructing conventional GIS coverage for each census. However, our need to compute rates meant that we needed to be able to link in counts taken from the Census of Population for use as divisors. For example, although the published Poor Law statistics generally include the total population of each union at the previous census, we were particularly interested in numbers of "able-bodied male" (ABM) paupers, and for them, a more appropriate divisor is the number of working-age males, which is available only in the census reports. Similarly, understanding this ABM rate requires knowledge of the occupational structure of each area, and especially numbers of farm workers, which again means linking in census data. Therefore, it was essential both that we generally treat unions and registration districts as one and the same, and that we clearly identify the exceptions. The subsequent development of this historical GIS meant that most of the usage was with registration district data, both census and vital registration, but in the last two years, much more poor law data has been computerized.

In building the original GIS, slightly separate time-variant coverage was created for unions and registration districts, but the design of the AUO was more rigorously focused on defining entities. Work on these geographies began with the listings of Poor Law units in Youngs' work [22,23]. These distinguish between the generality of unions, created by the 1834 Act, those created by earlier legislation, and those parishes that were large enough to have their own workhouse and officials. Therefore, we computerized these lists, defining each unit as being part of the relevant union/registration county, but as we knew that these units would also be used for registration and census data, we gave them the type PR_DIST, or 'Poor Law/Registration District', and similarly gave the counties the type PR_CNTY. We also included Youngs' information on precise legal status using the status table, which was already discussed, and the last four status values listed in Table 1. Then, the hierarchical

relationships between individual parishes and these units were added from Youngs' data as the parish entries were computerized.

Youngs makes no reference to registration districts, but our statistical database already contained very extensive transcriptions of registration and census statistics. We began by matching listings of districts and registration counties from 1911 to the AUO, giving those PR_DISTs that matched the additional status of RegD, and also adding any variant names used in the statistical reports. In all the subsequent linking-in of statistical data, census and registration data were only matched to units with RegD status, and Poor Law data were only matched to units with another status; in the process, the status data was further refined, and dates were added, as well as additional variant names. The final result is as shown in Table 1. There are 724 units of type PR_DIST, of which 670 were at some point registration districts, 644 units that were Poor Law unions, plus the three more specialized kinds of Poor Law unit. Note that Poor Law parishes are in fact exactly the same unit as the ordinary parishes that we also hold and identify as their only component area, but statistical mapping is greatly simplified by this split. We have had to make these trade-offs repeatedly between historical precision and the needs of statistical mapping, but different trade-offs are appropriate to different geographies, depending on how much statistical data is available.

**Table 1.** Status values defined for unit type PR_DIST.

| Status ID | Name | Meaning |
|---|---|---|
| RegD | Registration District (670 units) | Area covered by a Superintendent Registrar under the 1836 acts. Unlike unions, registration districts could be divided into registration sub-districts, smaller groupings of parishes in rural areas, or sub-divisions of parishes within cities. |
| PLU | Poor Law Union (644 units) | Grouping of parishes under the 1834 Act. |
| PLPar | Poor Law Parish (51 units) | Following the 1834 Act, some urban parishes had large enough populations to employ their own officials, and were not grouped into unions. |
| Inc | Incorporation (32 units) | Groupings of parishes created by earlier private poor relief acts between 1696–1795. |
| GilU | Gilbert Union (8 units) | The Gilbert Act (1782 (22 Geo. III, c. 83)) enabled parishes to unite to build a common workhouse. |

Fully explaining how this works in practice requires an even more specific case study, based on the major textile manufacturing center of Bradford, in Yorkshire, and its surrounding villages, as shown in Figure 2. Bradford was one of the northern areas that faced visible local opposition to the creation of the Bradford Union, with two riots in late 1837 at the initial meetings of the Board of Guardians [32]. In mid-19th century census and registration listings, for example in 1861, this area is covered simply by the Bradford district. However, 1861 poor law reports list two units: 499a North Bierley and 499b Bradford. Today, Bierley is a minor settlement just south of Bradford, but the 1887 listing noted above defines North Bierley Union as having been created in 1848 and consisting of a ring of 16 parishes entirely surrounding Bradford.

However, this was further complicated by boundary changes. We know that the pre-1892 Bradford Registration District (unit 10585545) covered the whole area identified, as indicated by the outermost black line. Within this early iteration of Bradford Registration District were two Poor Law unions, with Bradford (10139715) being the urban core created in 1837, while the North Bierley Poor Law Union (10144190) that was created in 1848 completely surrounded it, covering the adjacent rural area.

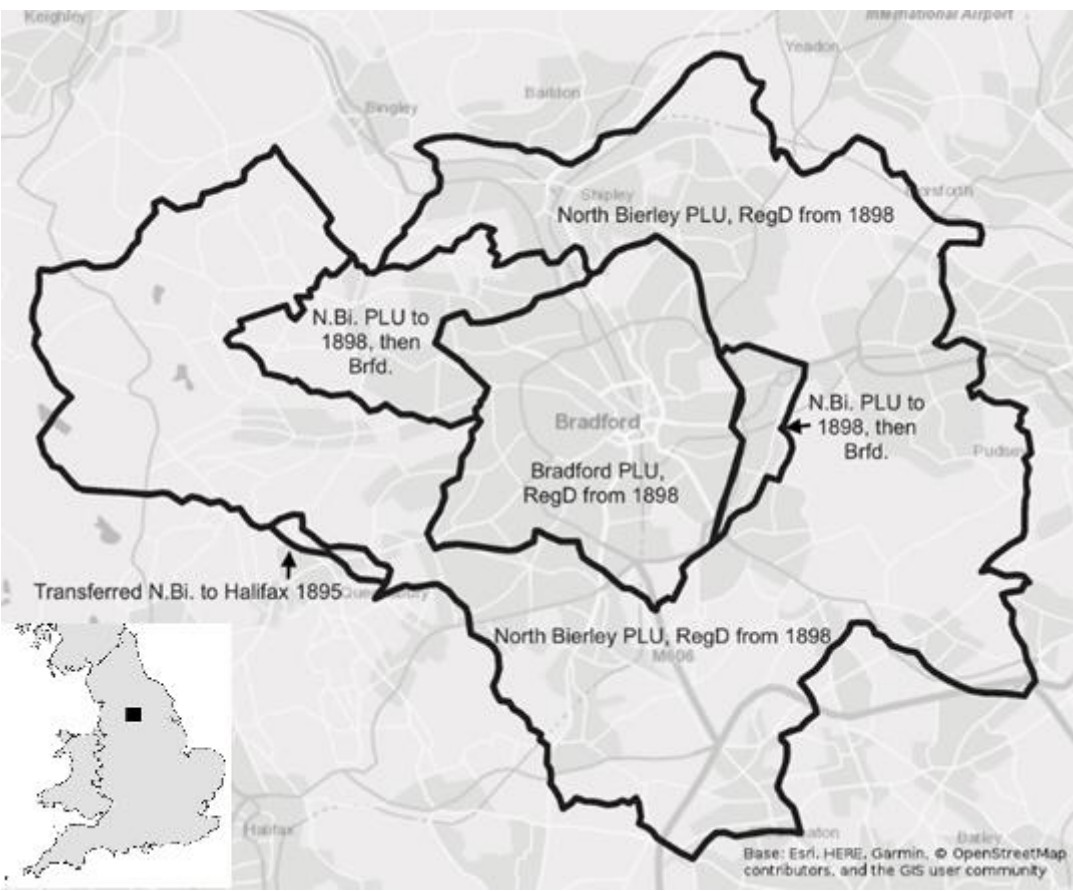

**Figure 2.** 'Footprints' for Bradford and North Bierley as Poor Law Unions and Registration Districts.

On 1 January 1898, the original Bradford Registration District (unit 10585545) was abolished. The whole geographical area was divided up to create two new registration districts, which were called North Bierley and Bradford. These two new registration districts corresponded in area to the pre-existing and continuing Poor Law unions. The registration district of North Bierley (10144190) lost small areas to the Halifax Registration District (10102560) during the boundary changes in 1895 (population 2777) and 1899 (population 446 — not depicted in Figure 2 due the small area and large map scale), and a significant area to the new smaller Bradford Registration District (10139715) on 1 October 1898 (population 13,386) as the city expanded. An additional change in 1898 was that the Bradford Poor Law union became a Poor Law parish as it became large enough to operate independently. This was caused by the abolition of six adjoining parishes and their inclusion in the Bradford civil parish to enlarge it on 25 March 1898.

This research into the exact composition of historical statistical areas is clearly intricate, and far more time consuming than defining ontology schemas or database architectures, but it is essential for accurate data analysis. Clearly, we cannot link 1861 census data for the Bradford district to Poor Law data for the Bradford Union. One solution would be to merge the two unions, but in this and many other cases, further research enabled us to use 1861 data on the age structure of lower-level registration sub-districts to compute exact numbers of working-age males for the two unions: listings of component parishes show that the North Bierley Union was coterminous with a set of nine sub-districts within the Bradford District, and the Bradford Union with the remaining four. Figure 3 is one final result from this work. It shows firstly a complete coverage without any holes for the whole of England and Wales, and secondly the high level of poor relief in the grain-growing area of eastern and southern England, even at harvest time.

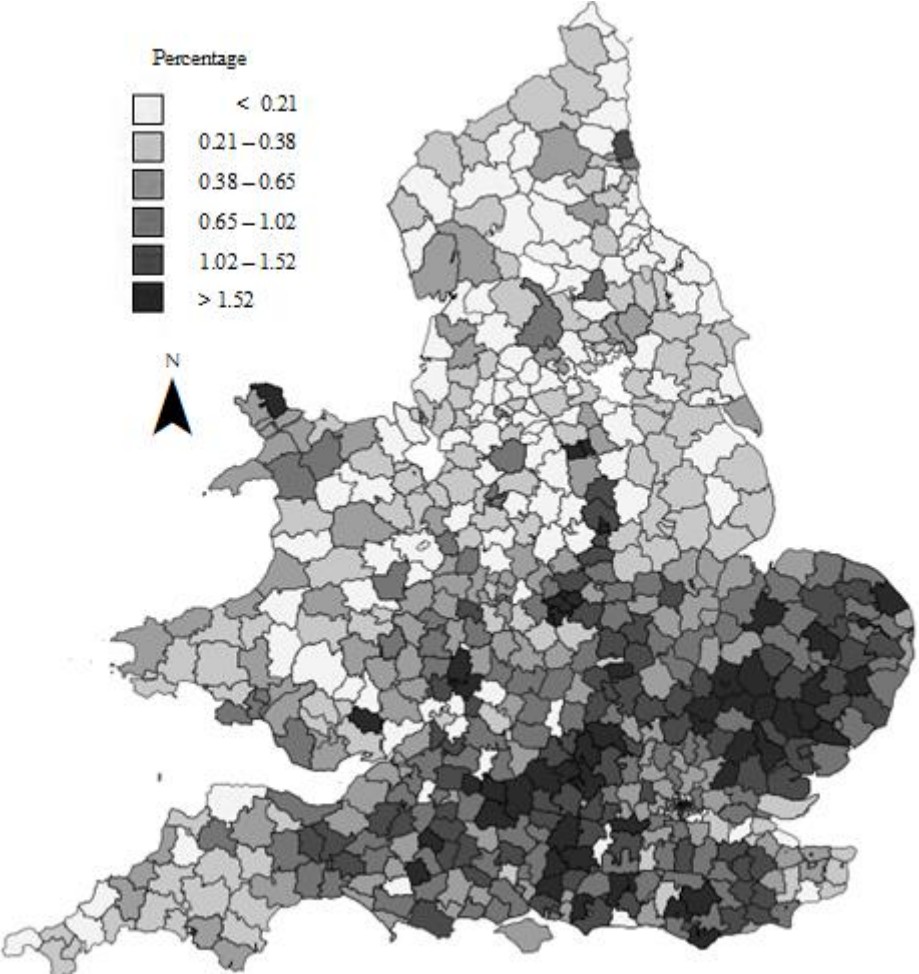

**Figure 3.** Able-bodied males on poor relief on 1 July 1861 for England and Wales, from the Poor Law board returns to Parliament, as a percentage of all males aged 15 to 64, as recorded by the 1861 Census of Population.

## 4. Defining States and Nations

Poor Law unions and registration districts were defined under the laws of England and Wales, and through the work of London-based officials; and most local administrative units were and are ultimately defined by national governments. However, any overall ontology of territorial administrations must also include those governments, and while they are better known than individual registration districts, both their existence and their boundaries are much more likely to be contested; here, contestation can be by scholars, diplomats, or armies. Felix Driver spoke of the "irreducible complexity and inter-relations of different forms of power relations and their effects" within and between states [33].

In some senses, sovereign states define themselves, but they are further defined through their relationships with other states, and with international organizations such as the European Union and the United Nations. As discussed below, there are states such as Somaliland that are purely self-defined, but their existence is inherently tenuous. Even where the existence of a state is unquestioned, its precise boundaries may be contested. Some boundaries very clearly follow physical features, such as mountain ranges, but others have had to be negotiated and then documented through international treaties. Historically, borders running through sparsely populated areas have not always been precisely defined. In some non-Western cultures, state-like entities were arguably defined more through ethnicity than territory, and relatively unconcerned with precise boundaries, especially where populations were migratory. Of course, in other situations, states have been very concerned about precise boundaries

but disagreed, leading to co-existing alternative lines, and sometimes to 'front lines' that moved as two armies fought.

The US State Department's Large-Scale International Boundaries dataset [34] (hereafter 'LSIB') is generally recognized to be the most topographically accurate global record of modern international boundaries. However, as discussed below, it incorporates certain positions taken by the US government that are not universally agreed, and it does not claim to be a systematic listing of the formal names of countries. Therefore, we also drew on listings published by the UK Foreign and Commonwealth Office that do include those formal names [35], including the membership list of the United Nations, which includes dates of admission [36], the US Central Intelligence Agency's World Fact Book [37], and the countries included in the World Bank's World Development Indicators dataset [38].

The LSIB defines 278 named multi-polygons, but these include 19 areas labeled as 'disputed' between two or more countries, and 62 'dependencies', which are identified by having the country they are dependent on named in brackets; for example, the Falkland Islands are labeled as both a British dependency and as disputed. This leaves 197 'countries', but at least six are somewhat problematic. In addition, there are states that claim existence but are not in the LSIB [39]. These ambiguous cases are listed in Appendix A.

If there is this much complexity in mapping the current set of states, how best to create a historical record? Many of the sources are straightforward. 'Independence Documents of the World' [40] is an extensive compilation of 'self-assertions' by states, but the most important sources are international treaties, and the many maps therein. Twentieth-century boundary changes are summarized in a series of International Boundary Studies published by the US State Department [41]. For the 19th century, two key sources are the multi-volume 'The Map of Europe by Treaty' [42] and 'The Map of Africa by Treaty' [43], which were both compiled by the UK Foreign Office's long-serving librarian. These are paralleled in the 20th century by 'The Map of Mainland Asia by Treaty' [44] and 'The Map of Latin America by Treaty' [45]—the last just an article, reflecting a simpler history.

Our immediate concern is mainly with Europe and to some extent its colonization of Africa, so the Hertslet compilation is a central source, with many maps. Obviously, they present the British government's positions and knowledge, but Britain in the 19th century, similar to the United States for much of the 20th, was a 'great power', involved in almost all international negotiations and a party to the resulting treaties. For example, in 1821, Britain was a signatory to the Convention on Cargoes on the [river] Elbe. There is still an issue of how far we can include unrecognized states, which for a time governed territory without being a party to treaties, but one element of a functioning government is a postal service, so one additional source is Stanley Gibbons' stamp catalogue [46].

One way to represent this data within the AUO would be to define 'STATE' as a unit type, and then enumerate all the sovereign states with the dates at which they came into existence, or ceased to exist. However, in practice, most sovereign states did not suddenly come into existence, but rather became independent, and there are sometimes gradations of independence. These issues are particularly important when the aim is not just to enumerate states but to provide a framework for also recording lower-level units down to the parish/village level. The resulting complexities have led to us twice revise how such entities are represented in the AUO.

The original AUO was limited to the British Isles, but still needed to define entities to hold 'national totals'. Therefore, it included five 'states' and five 'nations', as shown in Figure 4. Note that it is very unclear whether England has any legal existence, as unlike the other nations, it lacks a government other than that of the UK. Ireland has to appear twice, and each Irish county was given an 'IsPartOf' relationship with both Irelands. If the system had been taken back to before 1707, similar issues would arise with Scotland, and they could of course arise in the future.

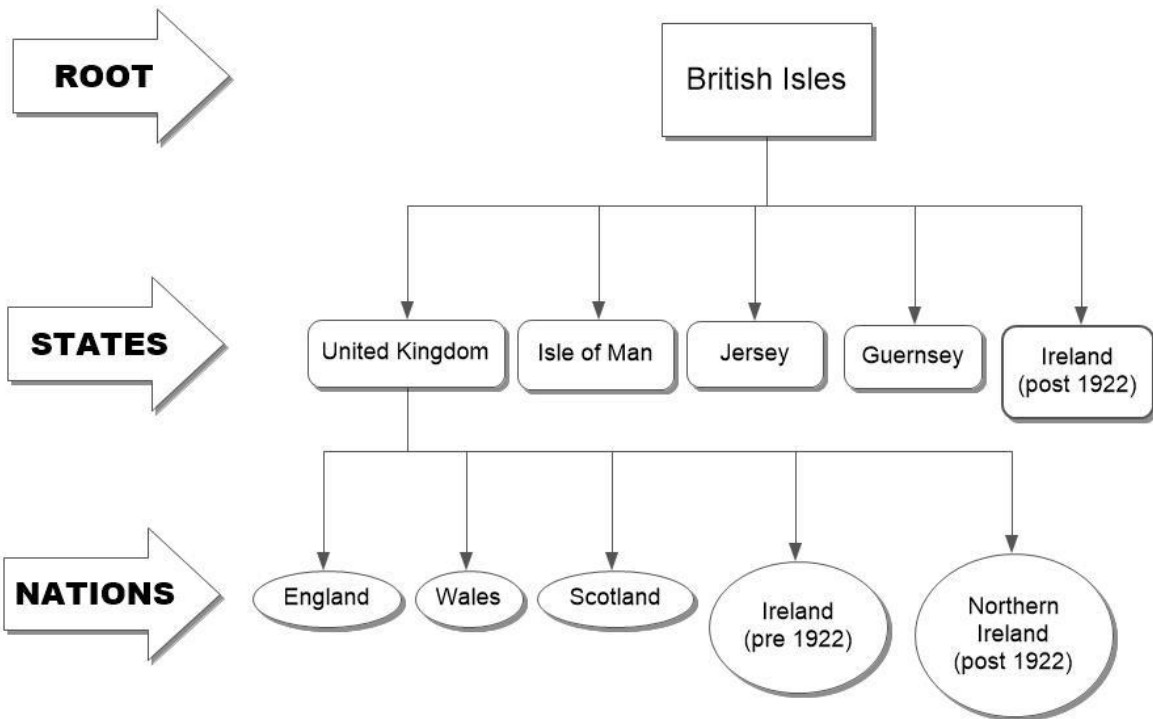

**Figure 4.** Original Incarnation of AUO Structure, 2003–2008.

The EU-funded QVIZ project required expansion across Europe, although the only additional detailed content was for Sweden and Estonia. The former was unproblematic, as Sweden formerly ruled a wider area, but everywhere in modern Sweden has always been in Sweden. However, representing Estonia was more challenging, as it was independent between 1918–1940, and since 1991, but at other times was ruled by the Polish–Lithuanian Commonwealth, Sweden, Russia, Germany, and the Soviet Union, while throughout its history, it was sub-divided into a relatively stable geography of counties (Maakond), and then parishes. Given that units can only be created once and abolished once, the immediate solution was to borrow the NATION type already defined for British units, link all the lower-level units to an Estonian nation with a continuous existence, and then make the nation part of either those other empires or of two Estonian states covering the periods of independence.

This representation, with three separate Estonias, matched our Estonian colleagues' sense of their own history, but would not work well if and when non-British historical statistics were added to the system. More recently, these definitions have further evolved, partly as a result of collaboration with the Collaborative for Historical Information and Analysis (CHIA) project in the US (http://chia.pitt.edu/index.php). The two separate types, STATE and NATION, have been replaced by the single type—COUNTRY—and the separate status table used to identify periods of dependence and independence.

Table 2 shows the current set of status values associated with the COUNTRY unit type, but this is still under development. The first four statuses are broadly derived from the LSIB data, the two most important being 'dependency' and 'sovereign': independence is now recorded by the end dates on a 'Dep' status value and on an AdministeredBy relationship, and the start date on a 'Sov' status value. Establishing precise relationships can be complex. For example, the LSIB data identify the Cook Islands, Niue, and Tokelau as dependencies of New Zealand, but van der Beek provided a detailed legal statement [47].

**Table 2.** Status values for countries.

| Status | Meaning | Notes |
|---|---|---|
| Sov | Sovereign State | For modern countries, this must be undisputed |
| Dep | Dependency | Administered by another country |
| Disputed | Disputed | Disputed territory |
| Joint | Joint | Jointly administered by two or more countries, usually under treaty |
| CrProt | Crown Protectorate | Because we still include British islands |

Even so, this approach is clearly simplistic and would not satisfy an international lawyer, so we also allow for more precise values. The current example is 'Crown Protectorate', which is the precise legal status of the Isle of Man, Jersey, and Guernsey (which includes the other Channel Islands of Alderney and Sark): these 'islands in British Seas' are 'possessions' of the British crown, but are not legally part of the United Kingdom, or consequently of the European Union (highly significant for tax purposes). These units simultaneously also have the broader 'dependency' status.

This will enable us to relax the current simplistic assumption that all countries are completely independent or 'dependencies'. In the 20th century, this essentially administrative conception of the state dominated, but in the 19th century, the patrimonial and feudal understanding of the state still mattered. Familial and other ties constrain sovereignty, and one practical consequence was the existence in Europe, on some counts, of 14 grand duchies. Norbert Elias related these institutional distinctions to the conflict between the central authority (Zentralgewalt) and the centrifugal forces (Zentrifugalkräfte) during the transformation from the feudal state based on personal ties to the administrative–legal territorial state [48,49].

Considerable work has already been done defining the countries of the world, and especially Europe, within the AUO, but detailed boundary mapping lies ahead of us. While the original historical work on Britain began by mapping boundaries from out-of-copyright maps published circa 1911, and then worked both forwards and backwards in time adding changes, the public domain LSIB mapping provides an accurate and up-to-date starting point from which we can work backwards. Although topographical accuracy is a concern, any international project cannot work to the same level of accuracy as is possible for a single country, so in the very long run, our historical boundary lines should be replaced by boundaries from national projects.

Therefore, the accuracy of dating and clear grounding in legal instruments are more important than absolute topographic accuracy. Thus, the very extensive mapping drawn from treaties and assembled into the Hertslet volumes on Europe and Africa are our most important source, which is to be supplemented by the original treaty maps held in the UK National Archives. Figure 5 is one example, showing one of the most important new divisions created in 19th-century Europe: the dismemberment of Poland by the Congress of Vienna, following the Napoleonic wars. Note the additional detail map on the right showing the boundary within Silesia.

Maps in treaties of course show positions agreed by the parties, so generally show resolutions to conflicts. Armed conflicts are mostly relatively short, with fluid front lines that are probably best ignored in a system spanning continents and centuries. However, the current LSIB contains much longer running contestations, often in places where until recently, resolution was not that urgent, such as the frontier between India and China in the Himalayas, and largely uninhabited islands that now matter because of the value of the surrounding fishing and mineral rights. Representing these within a GIS is clearly problematic, but we will build on the work of the SyMoGIH project and their Geo-Larhra system [50], as illustrated in Figure 6. This shows the territory of Luxembourg during the formation of the new Belgian state in 1830–39, and the different positions of Belgium and the Netherlands ('Pays-Bas'). As with attribution, it is difficult or impossible to track such information in a traditional desktop GIS, but relatively straightforward in a spatially-enabled database such as PostgreSQL.

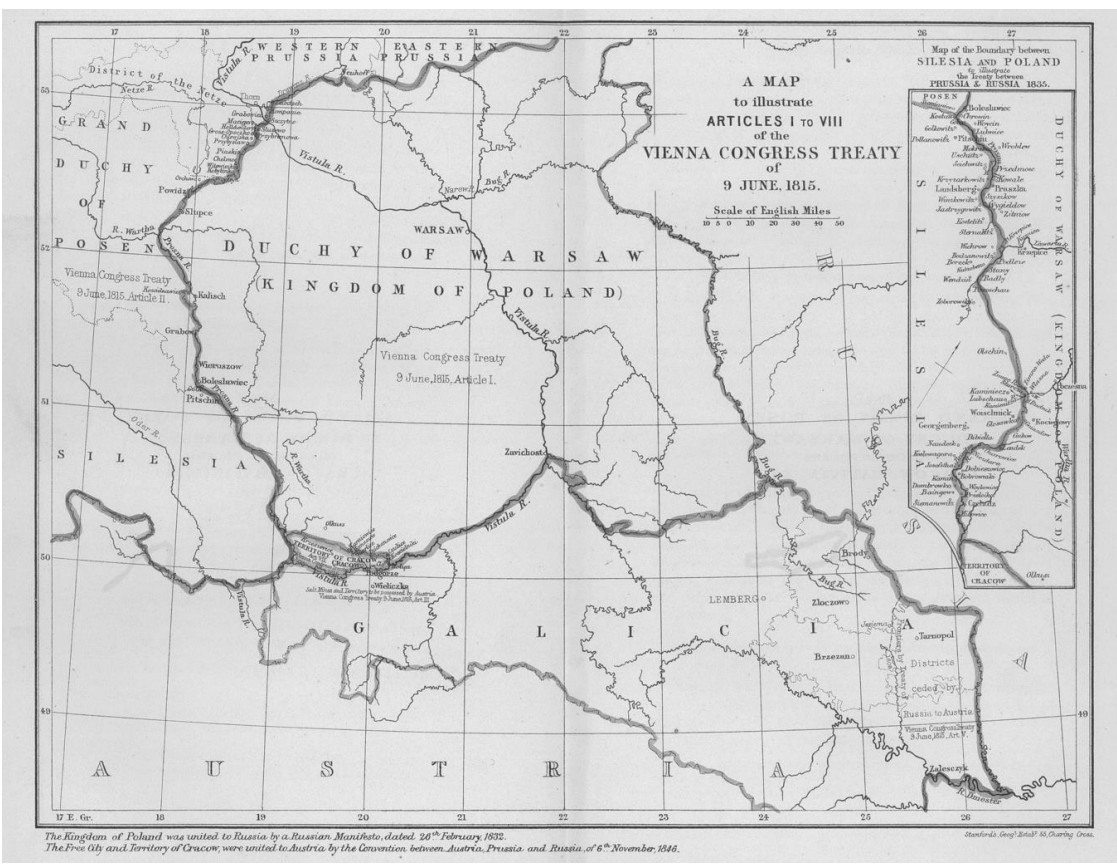

**Figure 5.** The Division of Poland by the Congress of Vienna, from Hertslet (1875, 1891).

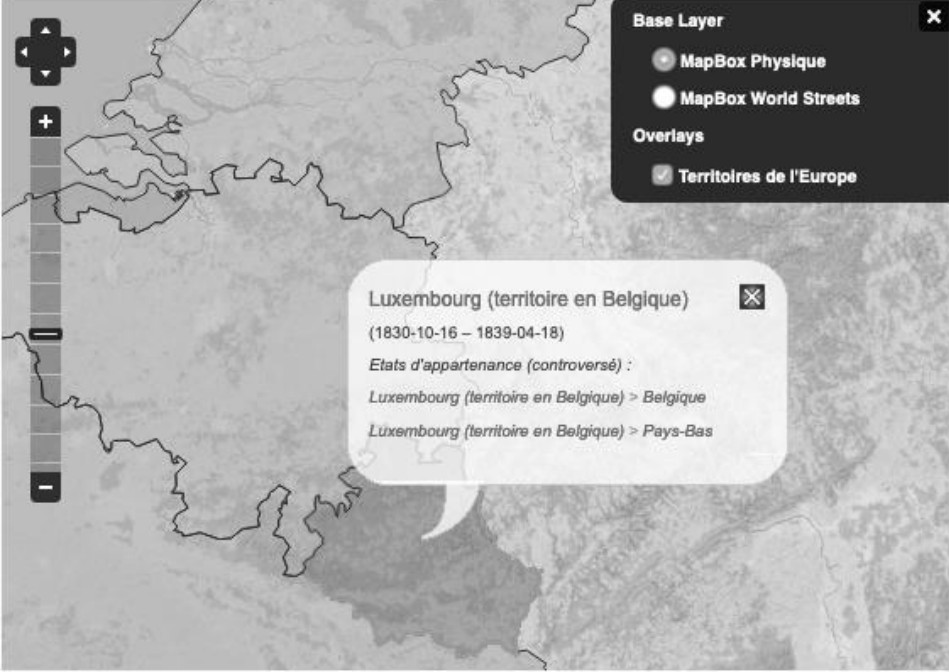

**Figure 6.** Luxembourg 1830–1839, as Represented in the Geo-Larhra System.

## 5. Conclusions

The research presented here is a work in progress, so far for 25 years, and one lesson is that a given body of historical information must take on new forms both as new demands are made on it and

as technology evolves: the object-relational software we now use to combine the semantic and the spatial barely existed when we started, and certainly lacked the performance that we now need, to support not only our research, but also a web site reaching a mass audience.

By now, we have mapped most of the administrative geographies that matter for Britain over the last 200 years, and the coverage presented here for the Poor Law and registration geographies is a very mature system that is rich in exact dates of changes and the other precise details of local history. However, we hope that we have shown that this is not mere cartographic antiquarianism, but an essential tool in the analysis of historical statistics, and the charting of long-run social and economic changes. In some senses, the most important output from this work is not statistical maps but rather a long-run time series, GIS technology enabling us to re-district data from the diverse reporting geographies used at different dates to a single constant reporting geography.

Work on multi-national historical GIS is far less developed, and for now, our main aim is to create a historically reliable online reference resource, which may one day become a framework for the integration of national historical GIS. We continue to be surprised that no one else seems to be attempting this task, but this partly reflects the failure of a whole series of proposals for multi-national projects, often because the need to include many national partners led to excessive budgets. Our priority is to create not a very large system, in terms of numbers of sub-national units included, but rather a system that is very thoroughly grounded in historical sources.

**Author Contributions:** Conceptualization and Methodology, Humphrey Southall; Investigation, Formal Analysis, Visualization and all Writing, Humphrey Southall and Paula Aucott.

**Funding:** This research received no external funding.

**Acknowledgments:** The construction of the GB historical GIS and the Administrative Unit Ontology since 1994 has been the work of a large and changing team, but we particularly note the work of Ian Gregory in creating the original time-variant GIS. We also acknowledge the contribution of our partners in the QVIZ project to internationalizing the system. Lastly, we thank Bogumił Szady of the Institute of History, Polish Academy of Sciences, and Francesco Beretta of the Laboratoire de Recherche Historique Rhône–Alpes, CNRS/Université de Lyon, for their contributions to Section 4 of this paper.

**Conflicts of Interest:** The authors declare no conflict of interest.

## Appendix A

There are at least six countries that are somewhat problematic:

- "Western Sahara" is recognized by the US government and the United Nations (UN), but was a Spanish colony that was annexed by Morocco as soon as the Spanish withdrew.
- Taiwan and Kosovo are functioning states, but they are not universally recognized, and are not members of the United Nations.
- The State Department data include polygons for "Gaza Strip" and "West Bank", but the United Nations includes these areas as the State of Palestine.
- Under the Antarctic Treaty of 1959, signatory states agreed to suspend all claims to specific territories, which had previously overlapped, but for example, the UK does still identify the "British Antarctic Territory", which is one of the suspended claims.

Additionally, there are states that claim existence but are not in the LSIB [51]:

- The Republic of South Ossetia and the Republic of Abkhazia are both breakaway areas of Georgia that are recognized by Russia but by few other countries. Similarly, the Turkish Republic of Northern Cyprus is recognized mainly by Turkey.
- The Republic of Artsakh and the Pridnestrovian Moldavian Republic are other former areas of the Soviet Union that are recognized only by non-UN members.
- Somaliland, the area of the former British colony, declared independence from Somalia in 1991, the remainder of that country having been an Italian colony. By most reports, its government is at least as functional as that of Somalia, but it is recognized by no other state.

- At the time of writing, Wikipedia lists 19 "rebel groups that control territory" [50], including the Islamic State and the Taliban, but the areas they control often change rapidly, and they are not necessarily providing "government".

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
