# Peer review of "Expressing History through a Geo-Spatial Ontology"

_ijgi, doi:10.3390/ijgi8080362_

Round 1
Reviewer 1 Report
This is a very professionally researched, organized, and written paper. The work presented here is valuable for a range of potential users, well beyond historians. Technically, it expands the reach of GIS to functions the current software was not developed to directly support, such as dealing with spatial change in the context of ambiguities, contradictions, contestations, errors, and other uncertainties in the definition of historical and contemporary statistical areas, and more generally, of territorial administrative units at any scale.
Of the two practical applications of the methodology outlined in the paper, the second and more speculative one, about states, countries and nations in the modern world, is the more interesting. The well-informed, thoughtful discussion does not suggest a solution to the ill-defined border problems at that global level, but demonstrates the challenges of pinning down many of these critical spatial entities in ways acceptable to all interested parties. We might have to agree that for particular territories at particular times, the 'correct' boundary definitions are a matter of perspective.
One thing I wish the authors had mentioned as part of a long-term research plan is the integration in their system of textual data from historical narratives that go beyond statistical information. Not only physical characteristics but also social and cultural ones are often behind the delimitation of administrative units, and may help in understanding the history of their changes over time.
Finally, I may have missed a reference to Harvard University’s “China Historical GIS” project, (http://chgis.fas.harvard.edu/), which also produced some innovative research on the general topic of this paper, also with an explicit temporal dimension.
I recommend that this manuscript be accepted in its present form.
Author Response
Reviewer 1 asked us to consider two points:
(1) I wish the authors had mentioned as part of a long-term research plan is the integration in their system of textual data from historical narratives that go beyond statistical information. Not only physical characteristics but also social and cultural ones are often behind the delimitation of administrative units, and may help in understanding the history of their changes over time.
We have already described the overall structure of our system at considerable length in the series of three papers in Historical Methods, and elsewhere; the present article has a more specific purpose. We are well aware that the detailed evolution of the boundaries of both North Bierley and Poland reflects diverse physical, social and cultural factors, but doubt that the International Journal of Geoinformation would publish narratives of the development of either “unit” which fully discussed these factors, i.e. at monograph length.
(2) I may have missed a reference to Harvard University’s “China Historical GIS” project, (http://chgis.fas.harvard.edu/), which also produced some innovative research on the general topic of this paper, also with an explicit temporal dimension.
We have had many productive conversations with the principals in the China Historical GIS, and have co-edited the “Placing Names” book with their technical lead. However, very little of a technical nature has ever been published by and about the China Historical GIS (or the US National Historical GIS). This makes it hard to cover in a literature review. We draw your attention to this list of “papers and presentations” on their web site, as well as to the results of searching Google Scholar for “China Historical GIS”:
https://sites.fas.harvard.edu/~chgis/papers/
Reviewer 2 Report
The paper is well written but the abstract should be improved, it should also be considered that the papers are usually written in the third person. I also find the work not very technical and therefore not very similar to my skills. In this sense, I would like to point out some paper that could be useful to analyze the spatio-temporal variations of the landscape from a more technical point of view.
· Statuto D.; Cillis G., Picuno P., 2019. GIS-based Analysis of Temporal Evolution of Rural Landscape: A Case Study in Southern Italy. Natural Resources Research. https://doi.org/10.1007/s11053-018-9402-7
· Cillis G., Statuto D.; Picuno P., 2019. Historical maps processed into a GIS for the assessment of forest landscape dynamics. Public Recreation and Landscape Protection - With Sense Hand in Hand... Conference Proceeding 2019, Pages 180-184.
Author Response
Reviewer 2 asked us to consider three points:
(1) the abstract should be improved:
We have reviewed the abstract, and made two small changes. However, overall it is an accurate summary of the paper. We are conscious that had we followed the general guidance provided by the journal on how the paper should have been structured, the abstract would also have had a particular structure. However, (a) we were invited by the guest editor to write a piece for this special issue, (b) when we read that guidance, we told him that we could not present our humanistic work within that rigid structure, so (c) we were instructed by him to ignore the structure and write as we thought appropriate.
(2) it should also be considered that the papers are usually written in the third person
Much of the paper concerns decisions we took and the reasons for them. Writing about our actions in the third person would be bizarre, and excessive use of the passive mode is bad style. We have sought the advice of the guest editor, who solicited this paper, and he told us to “do what you like”. We also note that the one paper already published for this issue, by Lan and Longley, consistently uses “we”.
(3) [It would] be useful to analyze the spatio-temporal variations of the landscape from a more technical point of view.
This seems to be an invitation to change our field of research to that of the reviewer. Part of the point of our paper is that historical research is distinguished from other areas of GIS not only or even primarily by the addition of a time dimension but by gaps in information, a great emphasis on textual information and a generally more humanistic approach. The papers the reviewer cites are essentially creating conventional time slices from historic maps, but in this present research our main focus is on contexts where maps do not exist, only texts, so you need gazetteers to have any kind of "geographical information".
Reviewer 3 Report
The paper presents a novel ontology to model the history of spatial entities and their attribute values.
The paper is relevant to the scientific community and bridges technical details and historical background information.
It is well-written, but some minor details should be changed prior to publication (see below).
My main concern is the list of contemporary counties in lines 322-346 which is a little off-topic and should go into the appendix. Also, some figures could be presented in a more appealing and interesting way.
I recommend a minor revision.
Line 4: Affiliations are identical, why listing them as 1 and 2?
Line 12: "GIS systems": please avoid this. Use "GIS" or "Geographic information systems" or "GI systems", also line 466.
Line 31: Another terminology remark: "GIS research": Here, "GIS" should stand for Geogr. Inf. Science, rather than systems. I would change to "research in Geogr. Inf. Science".
Line 49: "Say" sounds colloquial. Please use, "such as" or "e.g.", also line 74, 158, ... please correct throughout the paper.
Line 68: Somewhere here you could also cite the "Atlas of hist. county boundaries" in the U.S. (https://publications.newberry.org/ahcbp/index.html)
Line 149: "2002-3" means March 2003? Not sure if the month is relevant - I find this notation rather confusing.
Fig. 2: caption should be below figure. Also, could you add an inset map with the location of Bradford and North Bierley in a UK map?
Fig. 3: Please add in the caption which region is shown in the figure. Also, this figure would be even more impressive if the same variable was shown for a second (or more) point in time - if available - illustrating both, changes in the variable and in the aggregation boundaries. Also, is it possible to label some mayor cities as a reference unfamiliar with the shown study area?
Lines 322-346: While this list is interesting to read, it is too detailed, regarding the scope of this paper. Please consider moving this list to the appendix.
Line 417: Correct German would be "Zentrifugalkräfte".
Fig. 5: Could you overlay contemporary country boundaries on this map, to illustrate the changes for readers unfamiliar with Poland's shape ?
Author Response
Reviewer 3's general comments are mostly positive.
They do suggest that "the list of contemporary countries in lines 322-346 which is a little off-topic and should go into the appendix", and we have done exactly this.
They also suggest a reference to the US Atlas of Historical County Boundaries. This is not relevant to our review of administrative area ontologies, as for most of its existence this amazing project did not even use computers, but is very relevant to our revised discussion of historical sources so a reference has been added.
Reviewer 3 also provide a set of very detailed points:
Line 4: this was following the IJGI template, but has now been revised to list the institution once. Line 12: has been revised to be "GIS software", and similar changes have been made throughout. Line 31: Amended to be "GI research" as that is more accurate to our meaning, rather than geographical information science. Line 49: “say” has been replaced throughout. Line 149: These are years. Amended to be 2002-2003 etc. Line 417: This terminology comes from our Polish collaborator who is acknowledged at the end, so your reviewer may well be more familiar with the exact German term and we have revised the text accordingly. Figure 2: Caption moved below figure and inset locational map added. Figure 3: Caption amended to include region name. A key and north arrow added to make this figure clearer. We chose not to add another time slice example, but instead amended the text to explain more fully this figure emphasising the reasons for choosing this example. Given the dark hue of some areas of the map and the scale of the map, we felt the points the city labels would have marked would be too difficult to see so did not make this change. Our concern is with the broad geographical pattern. Figure 5: Applying a modern boundary layer would be problematic, and anyway we include this here as an example of an original historical source which should not be defaced.